# Biobased Cryogels from Enzymatically Oxidized Starch: Functionalized Materials as Carriers of Active Molecules

**DOI:** 10.3390/molecules25112557

**Published:** 2020-05-31

**Authors:** Antonella Caterina Boccia, Guido Scavia, Ilaria Schizzi, Lucia Conzatti

**Affiliations:** 1Institute for Chemical Sciences and Technologies-SCITEC “G. Natta”, CNR, Via Corti, 12, 20133 Milano, Italy; guido.scavia@scitec.cnr.it; 2Institute for Chemical Sciences and Technologies-SCITEC “G. Natta”, CNR, Via De Marini, 6, 16149 Genova, Italy; ilaria.schizzi@scitec.cnr.it (I.S.); lucia.conzatti@cnr.it (L.C.)

**Keywords:** cryogel, starch, NMR spectroscopy, morphology, drug release

## Abstract

Starch recovered from an agrifood waste, pea pods, was enzymatically modified and used to prepare cryogels applied as drug carriers. The enzymatic modification of starch was performed using the laccase/(2,2,6,6-tetramethylpiperidin-1-yl)oxyl TEMPO system, at a variable molar ratio. The characterization of the ensuing starches by solution NMR spectroscopy showed partial conversion of the primary hydroxyl groups versus aldehyde and carboxyl groups and successive creation of hemiacetal and ester bonds. Enzymatically modified starch after simple freezing and lyophilization process provided stable and compact cryogels with a morphology characterized by irregular pores, as determined by atomic force (AFM) and scanning electron microscopy (SEM). The application of cryogels as carriers of active molecules was successfully evaluated by following two different approaches of loading with drugs: a) as loaded sponge, by adsorption of drug from the liquid phase; and b) as dry-loaded cryogel, from a dehydration step added to loaded cryogel from route (a). The efficiency of the two routes was studied and compared by determining the drug release profile by proton NMR studies over time. Preliminary results demonstrated that cryogels from modified starch are good candidates to act as drug delivery systems due to their stability and prolonged residence times of loaded molecules, opening promising applications in biomedical and food packaging scenarios.

## 1. Introduction

Polysaccharides are natural and environmentally friendly polymers that have been used as starting materials for the production of a “new generation” of biobased materials because they are biocompatible, biodegradable, and nontoxic [1]. Native and modified polysaccharides, such as cellulose [2,3,4,5], hemicellulose [6,7], pectin [8,9], polygalactomannans [10,11,12], starch [13,14], and alginate [15] have been reported as promising matrices for producing bioaerogels via dissolution in water, retrodegradation, solvent exchange, drying via supercritical CO_2_, and air–liquid phase replacement [1]; and for producing cryogels via conventional lyophilization [10,11]. Cryogels are supermacroporous gel networks derived from the cryogelation of monomers or polymeric precursor gel matrices at the subzero temperature. Being lightweight and very resistant to the breakage materials and characterized by interconnected and open porous structures, large surface area, high mechanical strength, and ultralow dielectric constant, they appear suitable for a wide range of applications in several fields [16,17]. Additionally, considering they may be obtained through a simplest approach and in aqueous medium, cryogels are suitable and fit for diverse biological and biomedical applications, such as for drug release, immobilization of molecules and cells, and matrices for cell separation [18,19].

The purpose of this paper is to describe the synthesis of modified starch via enzymatic oxidation and the production of cryogels suitable as carriers of active molecules. The oxidation reaction was carried out by using fungal laccase, from *Trametes versicolor* and the mediator TEMPO (2,2,6,6-tetramethyl-1-piperidinyl-1-oxy radical) at a variable molar ratio [20,21,22]. Starch from pea pods (*Pisum sativum*) was used as feedstock derived from an agrifood waste. Starch is a polysaccharide with high molecular weight whose principal components are amylose and amylopectin. Amylose is a linear polymer of D-glucose units linked through α-(1 → 4) glycosidic bonds. Amylopectin has a branched structure through both α-(1 → 6) and α-(1 → 4) glycosidic bonds [23]. The combined use of laccase enzyme and the mediator TEMPO is a well-known method for the suitable oxidation of the primary hydroxyl groups to aldehydes [24,25]. The consequential formation of hemiacetalic bonds between the newly formed carbonyl and carboxyl groups and the free hydroxyl groups supports the creation of a crosslinked network responsible for the modified material behavior and allows the modulation of the material properties [25]. After the oxidation reaction, the modified starches were thoroughly characterized by mono- and two-dimensional solution NMR spectroscopy to determine the degree of oxidation and the nature of newly formed functional groups. Applying a conventional lyophilization process to modified starch, cryogels were obtained, and the morphology was investigated using atomic force microscopy (AFM) and SEM. The sorption/desorption capability of the cryogels was evaluated using caffeine in water, chosen as a “model drug” for the presence of functional groups in the molecular structure able to interact with those on starch polymer chains, thus favoring a slower desorption phenomenon. Moreover, this being a molecule that is a natural antioxidant by scavenging hydroxyl radicals, it is appropriate for biomedical applications to support the use of the proposed carrier [26]. Two different approaches of cryogel loading were evaluated, as reported: (a) adsorption of caffeine from the liquid phase (sponge cryogel); and (b) adsorption of caffeine from the liquid phase followed by a dehydration process (dry-loaded cryogel). Profile studies of release were then conducted by collecting a series of proton NMR spectra over time, thus showing encouraging preliminary results for the use as carriers. Synthesized cryogels, due to their properties, could have a wide range of promising applications in biomedicine (immobilizing biomolecules; capturing target molecules; for drug delivery; for wound healing), biotechnology, and bioseparation segments.

## 2. Results

### 2.1. Starch Oxidation

Purified starch from pea pod powder was enzymatically oxidized by using fungal laccase and the mediator TEMPO in mild reaction conditions and at a variable molar ratio (see Table 1), as described in Section 4.2. 

*Trametes versicolor* was chosen as a source of laccase because its carboxyl content was the largest compared to that of other enzymes [27] and because it is able to promote the oxidation of the mediator TEMPO in an unbuffered water medium. During oxidation, the mediator TEMPO is converted into an oxonium ion, able to selectively act on the primary hydroxyl groups present on pea pod starch (PS) chains, thus generating aldehyde groups. The chemical nature of PS derivatives was deeply investigated by mono- and two-dimensional NMR spectroscopy. Results were perfectly in line with the literature data, supporting experimentally the occurrence of the oxidation in low yield (10%) as observed for other polysaccharides, such as polygalactomannans [25] arabinoxylan and konjac glucomannans [10], for which a maximum of 12% of oxidation degree was reached. As a consequence, the resonances of the partially oxidized PS, in the proton spectrum of Figure 1, have small intensities overlapping with those of the native PS. The obtainment of aldehyde derivatives was confirmed by the appearance in the proton spectrum of characteristic resonances at 9.23 and 9.28 ppm in the ^1^H spectrum of Figure 1. Moreover, ^1^H chemical shifts of partially oxidized PS were assigned and are listed in Appendix A according to available literature data and two-dimensional experiments. Some peculiar signals in the proton spectrum were very close to those reported for acetylated starches [28,29] and oxidized polygalactomannans [24,25].

In the ^13^C spectrum of partially oxidized PS in Appendix A, the signals of produced acid residues are scarcely detectable because of the low yield of the oxidization reaction and especially the absence of NOE signal enhancement together with a long relaxation delay. In addition, the carbonyl signal expected in the 190–200 ppm region in the ^13^C spectrum cannot be easily detected, thus indicating that the aldehyde groups are hydrated or forming hemiacetals with the hydroxyl groups [30]. Nevertheless, the formation of a carboxyl carbon derivative for the modified PS was confirmed by the HMBC spectrum in Figure 2; this is because of the major sensitivity of the experiment along the proton dimension. Analyzing in detail the carboxyl region in Figure 2 (at 170–190 ppm), it is possible to observe a cross peak at 177.9 ppm referring to a *^3^J_CH_* correlation of a proton at 4.1 ppm with the carboxyl atom. This cross peak was assigned to the newly formed carboxyl group at the C6 position of the modified PS correlating through three bonds with the H4 proton. The enhanced technique also allowed easier detection of minor peaks attributable to the partially oxidized PS. Indeed, signals related to hemiacetalic derivatives were revealed from the HMBC analysis (at ca. 80–90 ppm), in agreement with literature data [25,30,31] and confirming the ^13^C NMR data. These derivatives can be justified from the chemo-enzymatic oxidation mechanism that supports the creation of a crosslinked network between the newly formed carbonyl and free hydroxyl groups. These groups are able to form intra- and inter-chain hemiacetalic bonds that are finally responsible for the modified material behavior.

Finally, the entire spin system was verified by 2D ^1^H–^1^H double quantum filtered-total correlation spectroscopy (TOCSY) experiment reported in Appendix A.

### 2.2. Cryogels

The final recovered cryogels from the modified PSs, obtained as illustrated on Section 4.2, showed different structures depending on the mediator TEMPO/laccase ratio in Table 1; therefore, samples A and B both showed fragile textures as fluffy and easy water dispersible matrices (sample B in Figure 3a); by contrast, sample C evidenced a compact and reinforced structure, as shown in Figure 3b.

#### Cryogel Morphology

SEM analysis on sample C in Figure 4 shows a highly porous structure, with the dimension of pores appearing to be not homogeneous in the whole fragile fractured surface. Indeed, areas with both smaller and bigger pores are observed. Moreover, nanometric holes are visible within some walls.

The morphological analysis carried at higher magnification by AFM in Figure 5 confirmed the presence of pores heterogeneously shaped and sized, of which the smallest ones range from approximately 200 nm to a few microns. Grain morphology of the flat areas (inlet of Figure 5b) was also evidenced.

This morphology with heterogeneously shaped pores is similar to that reported for superabsorbent aerogels obtained from cellulose nanofibrils [32,33] and hydrogels for tissue regeneration [34]. The presence of pores heterogeneous in shape and size together with nanometric pores in the walls could be useful for tuning the absorption/release of active molecules, such as caffeine.

## 3. Discussion

### Cryogels as Carrier

The ability of the partially modified PS to act as a delivery system of active molecules was investigated by solution NMR spectroscopy, as this technique is not limited to specific classes of compounds or functional groups but can be extended to the determination of all hydrocarbon compounds and is very useful for the analysis of mixture. Moreover, together with the release profile, it is possible to observe when the cryogel starts to dissolve. Finally, it is a quantitative and non-disruptive technique.

To test the cryogels’ ability to act as drug carrier, caffeine was chosen as model molecule because of the affinity of its functional groups with those of modified PS and because it is a well-known antioxidant and pro-oxidant and has reinforcing properties [35]. All samples in Table 1 were loaded with caffeine by immersing each of them into an Eppendorf tube containing caffeine in water, as illustrated in Section 4.3. (route (a)), and finally, samples A and B were discarded because of their fragile texture and easy water solubility; thus, cryogels from sample C conditions were used like carriers due to their compact and stable structure. Successively, the uploading capacity was evaluated for both samples. The main difference between the two loading methods is in the wet or dry use of the cryogels. For wet uses, the idea is to enhance the dissolution rate of poorly water-soluble drugs, thus increasing the therapeutic effects linked to drug availability. In the case of dry-loaded cryogels, wound healing applications are considered. This cryogel may be able to generate a wet gel at the wound site when an exudate is present, thus avoiding perilesional skin damages.

Firstly, for sponge cryogels from route (a), the release profiles of adsorbed caffeine were evaluated by collecting a series of proton NMR experiments acquired as detailed in Section 4.5. Direct quantification of released caffeine was done through a proportional comparison between the signal of the internal standard (TMS at δ = 0 ppm) and a selected signal of caffeine (e.g., a methyl signal at δ = 3.32 ppm,) considering that in a proton spectrum, the area of the signals is directly proportional to the number of protons present in the active volume of the sample [36]. Milligrams of released caffeine were determined by applying Equation (1) and then plotted vs. contact time in Figure 6 (considering cumulative milligrams over time). Data analysis for the sponge cryogel evidenced that the maximum of released caffeine was reached in the first 1560 min of elution, probably due to the quick release of caffeine from the surface of the PS material. Successively, for more prolonged contact time, the concentration goes down, and a more constant profile was reached probably because the caffeine entrapped in the core system was slowly released over time. After prolonged contact time, the cryogel starts to dissolve. Two stock solutions were analyzed to replicate data showing the same trend.

In the case of the dry-loaded cryogel from route (b), the procedure for the quantitative analysis of released caffeine was the same as described above. The plot of NMR data vs. contact time in Figure 6 evidenced no significant differences on the release profile for this material with respect to the sponge cryogel. The only valuable difference is attributable to the stability of this material that is more fragile and difficult to handle. In addition, in this case, two stock solutions were analyzed to replicate data showing the same trend.

These preliminary studies are encouraging and aspire to develop cryogels for specific applications. The sponge cryogels may contribute to the development of specific administration routes (topical, pulmonary, oral), thus shaping on demand, and on the “site” the drug availability. The dry-loaded cryogel will be useful for wound healing applications; it will mimic the natural tissue environment, and when loaded with antioxidant molecules, it can enhance antibiotic resistance during bacterial invasion [37].

## 4. Materials and Methods

### 4.1. Materials

Starch from pea pods (PS) in powder was generously provided by Dr. Marco Radice of Emsland Group and used after purification, carried out by dispersing it (10% *w*/*w*) in MilliQ water and ethanol in a 60/40 ratio by stirring at room temperature for 1 h. The material was recovered by filtration and then dispersed at 10% *w*/*w* in acetone and stirred for an additional hour at room temperature. After vacuum filtration, the material was dried at 50 °C overnight. The final yield was 94% (*w*/*w*).

Laccase from *Trametes versicolor* in powder form (0.5 U/mg as declared by Sigma-Aldrich), mediator TEMPO, caffeine, and all other chemicals were purchased from Sigma-Aldrich (Sigma-Aldrich, Darmstadt, Germany) and used as received. Deuterated water, (D_2_O 99.9%), and tetramethylsilane (TMS) for NMR spectroscopy were purchased from CortecNet (CortecNet, Les Ulis France) and used as received.

### 4.2. Oxidation Process and Cryogel Preparation

Purified PS was dispersed in milliQ water and stirred for 1 h at room temperature and then left overnight without stirring. Successively, a suspension of PS (10 mg) in water (2 mL) was prepared and stirred for 30 min at 30 °C, and then the mediator TEMPO and laccase were added in a molar ratio as reported in Table 1. The reaction mixture was stirred for 4 h at 30 °C and then kept at room temperature overnight. Finally, the enzyme was inactivated by placing the reactor into a boiling bath for 15 min.

The so-obtained reaction mixture, slightly viscous, was kept frozen at 8−0 °C for 18 h in cylindrical reactors and then lyophilized at −55 °C for 24 h (0.03 mbar). Recovered samples after lyophilization were stored at room temperature. Syntheses were duplicated to verify the reaction reproducibility. Furthermore, to verify the possibility of generating cryogels from the not enzymatically oxidized PS, the freeze-drying procedure was applied to a suspension of PS (10 mg) in water (2 mL) (sample D in Table 1) obtaining no cryogel.

### 4.3. Caffeine Adsorption in the Cryogel Structure

The adsorption of caffeine in cryogel was carried out following two different procedures, as follows: (a) for the sponge cryogel, a slice of 15 mm in diameter (5 ÷ 6 mg), obtained from sample C (in Table 1), was immersed into an Eppendorf tube containing caffeine in water (5 mM) for 90 min, then washed with 500 µL of water, and weighted to determine the “uploading capacity”; (b) the cryogel from route (a), after being charged with caffeine in water, was re-lyophilized, giving rise to the dry-loaded cryogel. Both procedures utilized the cryogel obtained from sample C condition in Table 1 because it is more stable in water with respect to samples A and B.

### 4.4. Studies on Sorption/Desorption of Caffeine in the Cryogels

The sorption/desorption capability of the loaded cryogel from route (a) was evaluated by immersing it into 700 µL of fresh deuterated water at regular interval time (respectively for 15, 30, 60, 180, 240, and 900 min as contact time) before acquisition of the NMR data. For all recovered solutions, proton spectra were recorded in quantitative conditions to evaluate the amount of released caffeine over time, and then this parameter was plotted as function of time considering the cumulative amount.

The sorption/desorption capability for the dry-loaded cryogel from route (b) was determined as detailed above by immersing it into 700 µL of fresh deuterated water at regular interval time, and proton NMR experiments were conducted on each solution. Finally, data were evaluated over time. Data were duplicated in both cases.

The experimental scheme illustrating the sorption/desorption procedure in the cryogels is reported in Appendix A.

### 4.5. NMR Spectroscopy

Mono- and two-dimensional NMR experiments were recorded on a 500-MHz Bruker DMX spectrometer, operating at 11.7 T, equipped with a 5 mm probe and gradient unit on z, and thermostated at 298 K (Bruker Biospin GmbH, Rheinstetten, Karlsruhe, Germany). The samples were prepared by dissolving 5.0 mg of oxidized starch into 700 µL of D_2_O at room temperature (The NMR data were acquired on modified PS before the lyophilization process, and no precipitate or solid materials was observed in the NMR tube.). As internal standard, 20 µL of a 0.7 mM TMS/water solution was added to each NMR solution before data acquisition.

Acquisition parameters for ^1^H experiments of enzymatically oxidized pea starch: 90° pulse 9.75 µs; PL1−2.2 dB; relaxation delay, 20.0 s. Spectral width, 8400 Hz; number of transient, 1024.

^13^C parameters: spectral width, 14 KHz, 90° pulse, 11.0 µs; PL1−1.3 dB with a delay of 10 s.

2D ^1^H-^1^H DQF-TOCSY (double quantum filtered-total correlation spectroscopy) was acquired with 256 experiments over 2 K data points and 256 scans each, with a mixing time of 0.09 s and a relaxation delay of 1.2 s.

The 2D ^1^H-^13^C g-HSQC experiments (gradient-heteronuclear single quantum coherence) were performed by applying a coupling constant *^1^J_CH_* = 150 Hz; data matrix 2 K × 256; number of scans: 128.

The 2D ^1^H-^13^C g-HMBC experiments (gradient-heteronuclear multiple bond correlation) were performed by applying a delay of 50 ms for the evolution of long-range coupling; data matrix 2 K × 256; number of scans 128; D1 2.00 s. Data were zero filled and weighted with a sine bell function before Fourier transformation.

Quantitative acquisition parameters for ^1^H spectra of caffeine solutions: 90° pulse, 9.75 µs; PL1−2.2 dB; relaxation delay, 40.0 s. Spectral width, 8400 Hz; number of transient, 1024. Data processing: exponential line broadening of 0.1 Hz was applied as resolution enhancement function; zero-filling to 32 K prior FT (TopSpin 4.0.6 software, Bruker Biospin GmbH, Rheinstetten, Karlsruhe, Germany). Spectra were referenced to the residual solvent signal of TMS at δ = 0 ppm, as internal standard.

For all experiments, spectra phasing and integration were performed manually, and the NMR spectra were processed using the Bruker TopSpin 4.0.6 software (Bruker Biospin GmbH, Rheinstetten, Karlsruhe, Germany).

Caffeine content was determined from the integral value in the proton spectrum by applying Equation (1) [36]:[mM]c = Ic/Hc [mM]st Hst/Ist(1)
where [mM]c is the millimolar concentration of caffeine; [mM]st is the millimolar concentration of the standard solution of TMS; Ic, Ist, and Hc and Hst are the integral value and number of protons generating the signals of caffeine and TMS, respectively. The ^1^H spectrum of caffeine with resonance assignment is reported in Appendix A.

### 4.6. Cryogel Morphology

The morphology and structure of the samples were assessed using scanning electron microscopy (SEM) performed on a Hitachi TM 3000 Benchtop SEM instrument (Tokyo, Japan) operating at 15 kV acceleration voltage. Observations were carried out on fragile fractures (in liquid nitrogen) of samples lyophilized and sputter-coated with gold.

Observations at higher magnification were carried out with a commercial AFM (NTMDT) model NTEGRA in tapping mode. For AFM measurements, the oxidized PS sample was fixed on a glass slide with a double tape.

## 5. Conclusions

Sustainable and renewable starch-based cryogels have been synthesized from enzymatically modified starch from pea pods, combined with conventional lyophilization. The nature of the functional groups derived from the oxidation reaction seems to play a crucial role in affecting the final behavior and properties of the synthesized materials. This work highlights the role of NMR spectroscopy as an analytical tool for material characterization and determination of the drug release profile of cryogels, allowing at the same time, to follow the material degradation process. The ability of the prepared cryogels to act as drug carriers might be useful in designing novel bio-inspired materials with promising application for wound healing and for specific administration routes in the pharmacological field. In the future, antimicrobial effects will be investigated to improve the performance of cryogels and to open the scenario on novel applications.

## Figures and Tables

**Figure 1 molecules-25-02557-f001:**
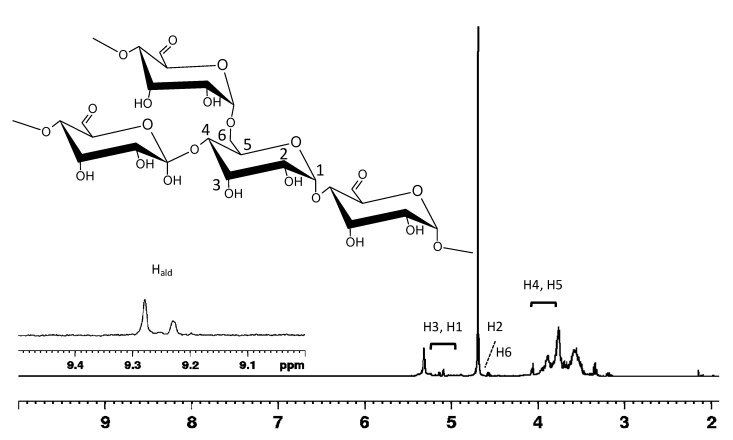
^1^H NMR spectrum of partially oxidized pea starch (sample C), recorded at 298 K, in D_2_O (for simplicity, only the numbering scheme of the amylopectin derivative is illustrated; the assignment refers to partially oxidized starch).

**Figure 2 molecules-25-02557-f002:**
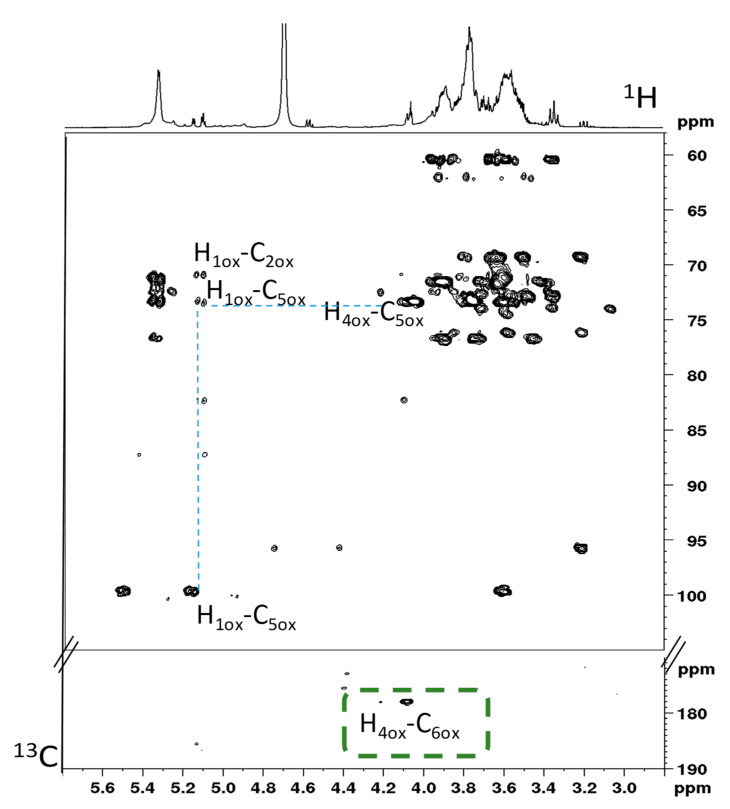
^1^H–^13^C HMBC spectrum of partially oxidized pea pod starch (PS) from sample C in D_2_O at 298 K. Some characteristic resonances are highlighted and refer to minor components from the oxidation process assigned with this technique.

**Figure 3 molecules-25-02557-f003:**
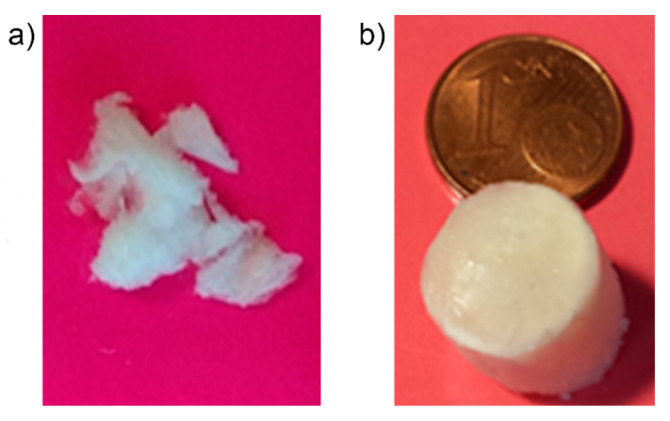
Cryogels from the modified PS as obtained from variable 2,2,6,6-tetramethyl-1-piperidinyl-1-oxy radical (TEMPO)/laccase ratio as shown in Table 1: (**a**) sample B; (**b**) sample C.

**Figure 4 molecules-25-02557-f004:**
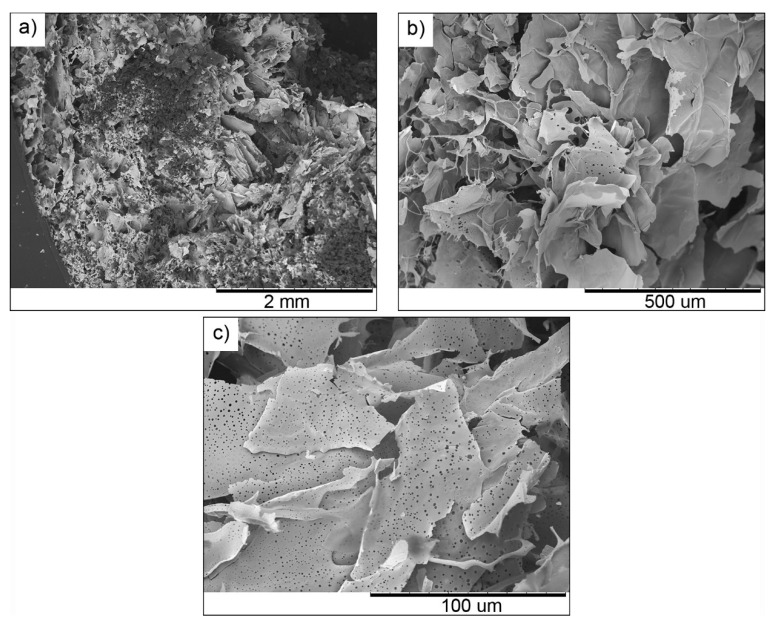
SEM images at different magnifications of the fragile fractured surface of sample C.

**Figure 5 molecules-25-02557-f005:**
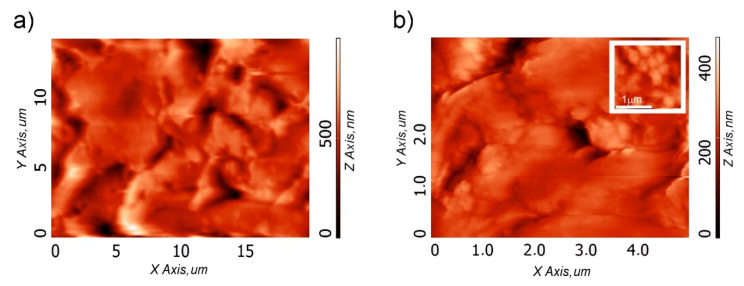
Atomic force microscopy (AFM) images of the sample C cryogel surface at: lower (**a**) and higher (**b**) magnifications. The inlet in (**b**) represents a magnification of the flat area.

**Figure 6 molecules-25-02557-f006:**
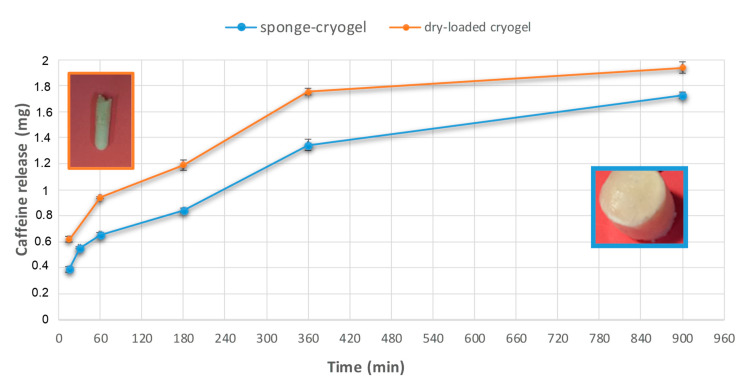
Percentage/mg medium values of the cumulative caffeine release versus time (standard deviation bars are reported) for the sponge cryogel and dry-loaded sample.

**Table 1 molecules-25-02557-t001:** Reaction conditions for the TEMPO-mediated laccase oxidation of PS in water.

Sample	TEMPO (mg)	Laccase ^1^ (mg)
A	1	4
B	1	40
C	10	40
D ^2^	-	-

^1^ Laccase was dissolved in 1 mL of water prior use. ^2^ Sample D was not enzymatically oxidized.

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
