# Peer review of "Biobased Cryogels from Enzymatically Oxidized Starch: Functionalized Materials as Carriers of Active Molecules"

_molecules, 2020, doi:10.3390/molecules25112557_

Round 1

Reviewer 1 Report

The idea of using starch-based cryogels as drug delivery systems sounds interesting; however the manuscript needs some additional data before being considering for publication. There are no error bars in any of the data showed. Why the authors decided to use NMR for the quantification of the released drug instead of HPLC? Also it is not clear why two loading methods were used and what would be the advantage of one method over the other.

Reviewer 2 Report

The manuscript molecules-794277 reports on preparation of freeze-dried enzymatically oxidized starch (laccase/TEMPO) and its loading with caffeine as an active compound. The manuscript reports an interesting approach towards starch-based drug-carrier system. The manuscript is of good technical quality. Conclusions are supported by data.

There are minor corrections to be made:

Methods:

  • -What is the physical state of starch after modification? Was any precipitate observed? Gel? Please specify.
  • Line 83: 10% acetone in which  solvent? water?
  • For sorption/desorption studies: were the data collected from two independent batches (line 123)? 

Data representation:

Figures  and 7 should be redrawn, two types of bars + values on top look redundant and nonprofessional. I suggest a simple plot. Error estimations should be added. Further, y-axis labels are inconsistent in these Figures. Suggest to plot relative amount released, not absolute (mg).

Reviewer 3 Report

In this paper, the authors present the modification of starch by a Laccase/TEMPO system to produce cryogels suitable as carriers of active molecules. They performed characterization of the materials from multiple perspectives (the morphology was characterized by AFM and SEM, the efficiency was studied by proton NMR studies etc.). They found that modified pea starch can act as a drug carrier based on its stability and prolonged residence times of loaded molecules.

Limitation:
1) Introduction has many mistakes: it not shows the importance of the topic (lacking clear motivation); nothing about drug delivery systems and about the applications of nanotechnology in this domain. In this section, the authors have written:

  • 2 from 4 paragraphs with 31 (too many; you're not writing a review article) from all 43 references about the well-known polysaccharides;
  • a very long paragraph about the aim of their study;
  • a last paragraph which is proper to be moved to the Methods.

The authors should discuss the advantages of their drug delivery system compared to other types (nano-gold particles, carbon nano-tubes etc.). Conclusion about this part: in the beginning, briefly describe the broad research area and then narrow down to your particular focus.

2) The authors should discuss more about why these procedures were selected, for example, what is the advantage of using caffeine?

3) Almost all Figures are about Sample C, but you have 4 samples. Comparative Figures are much more favorable.

4) The authors should provide more detailed discussion and interpretation on the materials characterization.

5) Only 9 from 43 references (~20%) are from the last 5 years. I reccommend to change the Introduction and to enrich the Discussion part with new studies from the literature (e.g. read M. Bakhshpour et al. Biomedical Applications of Polymeric Cryogels. Appl. Sci. 2019, 9, 553)

Round 2

Reviewer 1 Report

The manuscript seems much improved after revisions.

Reviewer 3 Report

All your changes are OK